# New Generations of Tyrosine Kinase Inhibitors in Treating NSCLC with Oncogene Addiction: Strengths and Limitations

**DOI:** 10.3390/cancers15205079

**Published:** 2023-10-20

**Authors:** Ilaria Attili, Carla Corvaja, Gianluca Spitaleri, Ester Del Signore, Pamela Trillo Aliaga, Antonio Passaro, Filippo de Marinis

**Affiliations:** Division of Thoracic Oncology, European Institute of Oncology IRCCS, Via G. Ripamonti 435, 20141 Milan, Italy

**Keywords:** TKI, targeted treatments, lung cancer, resistance, selective pressure

## Abstract

**Simple Summary:**

This manuscript focuses on improving the treatment of non-small cell lung cancer, with actionable gene alterations. The aim is to understand how the treatment with Tyrosine Kinase Inhibitors (TKIs) can be used and improved. Newer generations of TKIs have better results in controlling the disease and extending patient survival. These drugs also work better in the brain, which is crucial for patients with brain metastases. However, there are challenges. The use of newer TKIs may limit the role of older ones, and resistance to the drugs can emerge. The considerations from this manuscript suggest that understanding the biology of the tumor and the properties of the drugs could help develop new treatment strategies and ultimately benefit patients with this type of lung cancer.

**Abstract:**

Tyrosine kinase inhibitors (TKIs) revolutionized the treatment of patients with advanced or metastatic non-small cell lung cancer (NSCLC) harboring most driver gene alterations. Starting from the first generation, research rapidly moved to the development of newer, more selective generations of TKIs, obtaining improved results in terms of disease control and survival. However, the use of novel generations of TKIs is not without limitations. We reviewed the main results obtained, as well as the ongoing clinical trials with TKIs in oncogene-addicted NSCLC, together with the biology underlying their potential strengths and limitations. Across driver gene alterations, novel generations of TKIs allowed delayed resistance, prolonged survival, and improved brain penetration compared to previous generations, although with different toxicity profiles, that generally moved their use from further lines to the front-line treatment. However, the anticipated positioning of novel generation TKIs leads to abolishing the possibility of TKI treatment sequencing and any role of previous generations. In addition, under the selective pressure of such more potent drugs, resistant clones emerge harboring more complex and hard-to-target resistance mechanisms. Deeper knowledge of tumor biology and drug properties will help identify new strategies, including combinatorial treatments, to continue improving results in patients with oncogene-addicted NSCLC.

## 1. Introduction

Lung cancer remains one of the most prevalent and deadly malignancies worldwide, with non-small cell lung cancer (NSCLC) constituting the majority of cases diagnosed [1]. Over the past few decades, significant strides have been made in understanding the molecular underpinnings of NSCLC, leading to the identification of driver gene alterations that have, in turn, transformed the landscape of treatment for this disease. Among these remarkable advancements, the advent of tyrosine kinase inhibitors (TKIs) has emerged as a paradigm-shifting approach, offering newfound hope and extended survival to patients with advanced or metastatic NSCLC harboring specific genetic alterations [2,3].

The landscape of medical oncology has witnessed a revolution triggered by these targeted therapies, as they have the potential to halt the progression of the disease with greater efficacy and fewer adverse effects compared to conventional chemotherapy. Understanding the historical evolution and ongoing developments of TKIs is crucial for both clinicians and researchers in the field of medical oncology.

The inception of TKIs in NSCLC therapy began with the first generation of these agents. Initially designed to target the epidermal growth factor receptor (EGFR), this class of drugs showed unprecedented promise in a subset of NSCLC patients harboring *EGFR* mutations [4]. It was a watershed moment, offering personalized treatment options and substantially improving outcomes for these individuals. However, the first-generation TKIs, such as erlotinib and gefitinib, brought with them their own set of limitations, including the emergence of resistance mechanisms like the p.T790M mutation [5].

Recognizing the need for more potent and selective therapies, researchers swiftly moved forward in the development of subsequent generations of TKIs. These newer iterations promised enhanced specificity for their respective targets and a greater ability to circumvent resistance mechanisms [6].

Despite these remarkable advancements, the utilization of novel generations of TKIs is not without its complexities and challenges. Resistance remains a persistent issue, necessitating ongoing research into more effective treatment strategies [7,8,9]. Additionally, the optimal sequencing of these agents and their integration into the treatment landscape of NSCLC requires careful consideration (Figure 1). The identification of predictive biomarkers and the management of adverse effects associated with TKI therapy are also areas of active investigation [10].

This review manuscript will provide a thorough analysis of the strengths and limitations associated with the use of TKIs in NSCLC with oncogene addiction. We will explore the pivotal clinical trials that have shaped the current treatment paradigm and shed light on the emerging therapies currently under investigation. Furthermore, we will delve into the intricate biology underlying the potential strengths and limitations of these agents, unraveling the complex interplay between oncogenic signaling pathways and therapeutic interventions.

## 2. Novel Generations of TKIs for NSCLC in Clinical Practice

### 2.1. Efficacy

Since the advent of the first-generation TKIs, erlotinib and gefitinib, for the treatment of patients with *EGFR*-sensitizing mutations, an enduring paradigm shift towards precision oncology has guided the development of more potent and specific TKIs to overcome intrinsic and acquired resistance mechanisms responsible to treatment failure (efficacy results of novel generation TKIs in clinical practice are shown in Table 1).

Second-generation TKIs afatinib and dacomitinib improved clinical outcomes compared to platinum-based chemotherapy (PBC) and first-generation TKIs in patients with *EGFR*-sensitizing mutations [11,12,13,14,15,16]. Moreover, in a pooled analysis of the LUX-Lung trials, afatinib was also active in tumors with uncommon *EGFR* mutations, although the clinical benefit was lower in patients with de novo T790M and exon-20 insertion mutations [17]. However, after these molecules entered clinical practice, the occurrence of severe adverse events (AEs), mostly skin rash and diarrhea, due to the inhibition of wild-type EGFR narrowed the therapeutic window that was needed to effectively overcome acquired resistance mechanisms, especially the T790M mutation. To address this shortcoming, the third-generation osimertinib was developed to specifically target the *EGFR*-T790M mutation while retaining activity against initial activating mutations and selectivity over wild-type *EGFR*. Osimertinib received its first approval from the Food and Drug Administration (FDA) for the treatment of *EGFR*-T790M-positive NSCLC based on a 6-month improvement in progression-free survival (PFS) compared to PBC in the AURA3 trial and a hazard ratio (HR) for overall survival (OS) of 0.54 after adjustment for the high crossover rate in the study [18,19]. Subsequently, in the FLAURA trial, osimertinib outperformed first-generation TKIs in PFS (18.9 vs. 10.2 months, HR 0.46) and OS (38.6 vs. 31.8 months, HR 0.80), regardless of the T790M mutation, with better tolerability, establishing the role of first-line osimertinib as the gold-standard [20,21]. Despite these results, the emergence of resistance ultimately leads to treatment failure. Resistance mechanisms are highly complex and multifaceted, including the emergence of the C797S mutation, the loss of T790M, small cell lung cancer (SCLC) transformation, and *MET* amplification; thus, tumor biopsy upon disease progression should be considered whenever feasible to optimize treatment strategies [22,23].

Although relatively rare, accounting for 2–3% of cases, *EGFR*-exon 20 insertion mutations confer resistance to TKIs, requiring treatment with PBC. The oral TKI Mobocertinib was active and led to sustained responses in PBC-treated patients with *EGFR*-exon 20 insertion mutations. Based on these results, despite gastrointestinal and dermatological AEs hampering their clinical utility, mobocertinib was granted FDA accelerated approval [24,25]. However, the confirmatory trial EXCLAIM-2 ended prematurely in July 2023 as first-line mobocertinib monotherapy failed to improve PFS compared to PBC [26]. Therefore, it is still to be determined whether the approval will remain intact, particularly given the PFS improvement observed with the combination of first-line amivantamab, an EGFR-MET bispecific antibody, and PBC over PBC alone in the phase III PAPILLON trial [27].

In the realm of *HER2*-mutation-positive NSCLC, phase 2 trials have investigated the role of small molecule TKIs, pyrotinib and poziotinib. These agents showed only modest activity and severe gastrointestinal and cutaneous AEs, owing to *EGFR* inhibition, that hindered further development [28,29,30]. Current research is focusing on novel HER2-selective TKIs that lack activity against other HER/ERBB family members, aiming for enhanced activity and improved safety. Notably, in the phase 2 study DESTINY-Lung01 the antibody drug-conjugate (ADC) trastuzumab-deruxtecan (TDXd) yielded durable activity in previously treated patients and was generally well-tolerated, though interstitial lung disease (ILD) required prompt diagnosis and management [31], and the ongoing DESTINY-Lung-04 will determine its superiority over PBC as first-line (NCT05048797).

Following the approval of the first-in-class ALK-TKI, crizotinib [32], there was a compelling need for more potent therapeutic alternatives to overcome resistance. Owing to an inhibitory activity against several crizotinib or ceritinib-resistant *ALK* mutations, alectinib first improved PFS and intracranial objective response rate (ORR) in crizotinib-resistant patients compared to PBC, with an acceptable safety profile [33]. Subsequently, the ALEX trial established the superiority of first-line alectinib compared to crizotinib, with a 24-month PFS improvement (HR 0.32) and higher CNS activity (59% vs. 26%) [34,35]. First-line brigatinib also improved long-term outcomes over crizotinib in the ALTA 1L study and stands as a viable treatment option in this setting [36]. The third-generation lorlatinib, initially developed to overcome resistance mechanisms responsible for progression to second-generation TKIs, significantly improved PFS (HR 0.28) compared to crizotinib in the CROWN trial [37,38]. Treatment with lorlatinib was associated with an acceptable toxicity profile, as grade 3–4 AEs were mostly represented by altered lipid levels [38]. Interestingly, some compound mutations that confer resistance to lorlatinib might re-sensitize tumoral cells to crizotinib, making molecularly guided treatment a potentially valuable therapeutic strategy in some cases [39].

In the context of *ROS1*-fusion-positive NSCLC, crizotinib is associated with a median PFS of approximately 19 months [40], yet treatment failure and CNS progression generally occur within 2 years of treatment [41]. Lorlatinib demonstrated activity in crizotinib-resistant, *ROS1*-positive NSCLC in a Phase I-II trial, achieving an ORR of 35% [42]. Recently, in a pooled analysis of the phase I-II trials, ALKA-372-001, STARTRK-1, and STARTRK-2 entrectinib achieved an ORR of 67%, with a median duration of response (DoR) of almost 16 months, thereby supporting the choice of this agent for first-line treatment. Although generally well tolerated, severe AEs occurring at a low frequency, including cardiac and CNS AEs, need to be carefully monitored as they might require dose modifications in some instances [43,44].

Persistent efforts in the structural analyses of KRAS, a protein that has been historically deemed “undraggable”, paved the way for the development of *KRAS*^G12C^-selective inhibitors sotorasib and adagrasib, that, although not in the class of TKIs, are worth mentioning as they have been both approved for clinical use following at least one prior line of systemic therapy. Sotorasib demonstrated activity in the phase I/II CodeBreaK-100 trial, with an ORR of 41%. The most common AEs included diarrhea and elevation in transaminases. However, in the phase III CodeBreaK-200 trial, the PFS improvement was small compared with docetaxel (5.6 vs. 4.5 months, HR 0.66), and OS was similar in the two arms (10.6 vs. 11.3 months) [45]. Similarly, adagrasib achieved an ORR of 43% in phase I/II KRYSTAL-1 trial, with durable responses and grade ≥3 AEs in 45% of patients [46,47], while its efficacy as second-line compared to docetaxel and as first-line is currently under investigation (KRYSTAL-12, NCT04685135; KRYSTAL-7, NCT04613596).

Traditionally, *MET* gene alterations have been treated with crizotinib [48], and other multikinase inhibitors, with limited efficacy and significant toxicity. Selective MET-TKIs capmatinib and tepotinib have revolutionized the treatment landscape for patients with a *MET*-exon-14-skipping mutation, leading to high and durable responses in both previously treated (ORR 40–51%) and treatment-naïve patients (ORR 56–67%) in the GEOMETRY mono-1 and VISION trials, respectively [49,50]. Noteworthy, common AEs associated with these agents include peripheral edema, increased creatinine levels, and gastrointestinal events. Another specific MET-TKI, Savolitinib, was only approved in the People’s Republic of China in 2021 [51]. Notably, MET-TKIs have yet to receive approval for high-level *MET* amplification, although preliminary data warrant further investigation, and the ideal methodology for determining the level of amplification and appropriate cutoffs for treatment is still an active area of research.

In *RET*-fusion positive NSCLC, the RET-selective TKIs selpercatinib and pralsetinib earned approval in 2021, as strong clinical activity was observed in the phase I/II LIBRETTO-001 and ARROW studies, both in treatment-naïve (ORR 84% and 72%, respectively) and previously treated patients (ORR 61% and 59%, respectively) [52,53]. Common AEs for selpercatinib include hypertension and increased liver enzymes, while it is crucial to monitor the occurrence of ILD associated with pralsetinib [54,55].

For *BRAF*-V600E-mutant NSCLC, the combination of oral serine/threonine kinase inhibitors dabrafenib and trametinib obtained significant responses in both the first (ORR 68%) and second (ORR 64%) line in the phase II BRF113928 study, and it is considered standard of care [56]. Recently, encorafenib plus binimetinib showed comparable efficacy, with an ORR of 75% in treatment-naïve and 46% in pretreated patients, and this combination might emerge as a new therapeutic option [57].

Lastly, larotrectinib and entrectinib have received tumor-agnostic approval based on their efficacy in basket trials enrolling patients with neurotrophic tyrosine receptor kinases (*NTRK*) fusion-positive tumors. The phase I/II NAVIGATE trial and the pooled analysis of the STARTRK-1, STARTRK-2, and ALKA-372-001 studies demonstrated ORRs of 73 and 63%, respectively, in patients with *NTRK*-fusion-positive NSCLC. The incidence of grade 3–4 AEs, dose reductions, and discontinuations was low [58,59,60].

**Table 1 cancers-15-05079-t001:** Efficacy results in registrational trials for 2nd and 3rd generation TKIs and KRAS-inhibitors in clinical practice.

Oncogene	TKI	Registrational Trial	N° of Patients	Control Arm	Primary EP	Efficacy Results	CNS Activity in Patients with Evaluable Lesions
EGFR exon 19 deletions and exon 20 L858R	Afatinib	Phase IIB LUX-Lung-7 [14,61]	319	Gefitinib	PFS	Median PFS 11.0 vs. 10.9 months (HR 0.74; 95% CI, 0.57–0.95; *p* = 0.0178)	NA
	Dacomitinib	Phase III ARCHER 1050 [15,16]	452	Gefitinib	PFS	Median PFS 14.7 vs. 9.2 months (HR 0.59; 95% CI, 0.47–0.74; *p* < 0.0001)	-
	Osimertinib	Phase III FLAURA [20,21,62]	556	Gefitinib or Erlotinib	PFS	Median PFS 18.9 vs. 10.2 months (HR 0.46; 95% CI, 0.37–0.57; *p* < 0.001)	icORR 91% vs. 68% icDoR 15.2 vs. 18.8 months
ALK	Ceritinib	Phase III ASCEND-4 [63]	376	PBC	PFS	Median PFS 16.6 vs. 8.1 months (HR 0.55; 95% CI, 0.42–0.73; *p* < 0.00001)	icORR 72.7% vs. 27.3% icDoR 16.6 months vs. NE
	Alectinib	Phase III ALEX [34,35]	303	Crizotinib	PFS	Median PFS 34.8 vs. 10.9 months (HR 0.43; 95% CI 0.32–0.58, *p* = 0.0001)	icORR 81% vs. 50% icDoR 17.3 vs. 5.5 months
	Brigatinib	Phase III ALTA 1L [36]	275	Crizotinib	PFS	Median PFS 24.0 vs. 11.1 months (HR 0.48, 95% CI 0.35–0.66, log-rank *p* < 0.0001)	icORR 78% vs. 26% icDoR 27.9 vs. 9.2 months
	Lorlatinib	Phase III CROWN [37,38,64]	296	Crizotinib	PFS	Median PFS NR vs. 9.3 months (HR 0.28; 95% CI 0.19–0.41, *p* < 0.001)	icORR 83% vs. 23% icDoR NR vs. 10.2 months
ROS1	Entrectinib	Phase I-II ALKA-372-001, STARTRK-1, and STARTRK-2 [43,44]	161	-	ORR DoR	ORR 67.1% (95% CI 59.3–74.3) Median DoR 15.7 months (95% CI 13.9–28.6)	icORR 79.2% icDoR 12.9 months
MET Exon 14 skipping	Tepotinib	Phase II VISION trial [50,65]	111, 1L T+ (cohort C + A)	-	ORR	ORR 56.8% (95% CI, 47.0–66.1)	icORR 55% icDoR 9.5 months
			97, ≥2L (cohort C + A)			ORR 49.5% (95% CI, 39.2–59.8)	
	Capmatinib	Phase II GEOMETRY-mono-1 trial [49,66,67,68]	28 Treatment-naïve (cohort 5b)	-	ORR	ORR 67.9% (95% CI, 47.6–84.1)	iORR 67.9%
			32 Treatment-naïve (expansion cohort 7)			ORR 65.6% (95% CI, 46.8–81.4)	
			69 pretreated 2/3L (cohort 4)			ORR 40.6% (95% CI, 28.9–53.1)	iORR 40.6%
			31 pretreated 2L (expansion cohort 6)			ORR 51.6% (95% CI, 33.1–69.8)	
KRAS G12C	Sotorasib	Phase II CodeBreaK 100 [69]	174	-	ORR	ORR 40.7% (95% CI, 33.3–48.4)	icORR NR icDoR NR
		Phase III CodeBreak 200 [45]	345	Docetaxel	PFS	Median PFS 5.6 vs. 4.5 months (HR 0.66; 95% CI 0.51–0.86, *p* = 0.0017)	icORR 33%
	Adagrasib	Phase I/II KRYSTAL-1 [47,70]	116	-	ORR	ORR 42.9% (95% CI, 34.5–52.6)	icORR 42% icDoR 12.7 months
RET	Selpercatinib	Phase I/II LIBRETTO-001 [54,71]	69 Treatment-naïve	-	ORR	ORR 84% (95% CI, 73–92)	icORR 82% icDoR 9.4 months
			247 PPP			ORR 61% (95% CI, 55–67)	
	Pralsetinib	Phase I/II ARROW [55]	75 Treatment-naïve	-	ORR	ORR 72% (95% CI, 60–82)	icORR 78%
			136 PPP			ORR 59% (95% CI, 50–67)	
BRAF V600E	Dabrafenib/ Trametinib	Phase II BRF113928 [56]	36 Treatment-naïve (Cohort C)	-	ORR	ORR 63.9% (95% CI, 46.2–79.2)	NA
			57 Pretreated (Cohort B)	-		ORR 68.4% (95% CI, 54.8–80.1)	
NTRK	Larotrectinib	Phase I/II NAVIGATE [58]	20 NSCLC	-	ORR	ORR 73% (95% CI, 45–92)	icORR 63%
	Entrectinib	Phase I/II STARTRK-1; STARTRK-2; ALKA-372–001 [59,60]	22 NSCLC	-	ORR	ORR 63.6% (95% CI, 40.7–82.8)	icORR 67%

CI: confidence interval; CNS: central nervous system; DoR: duration of response; EP: end point; HR: hazard ratio; icDCR: intracranial disease control rate; icDoR: intracranial duration of response; icORR: intracranial ORR; NA: not available; NE: not estimable; NR: not reached; NSCLC: non-small cell lung cancer; OR: odds ratio; ORR: objective response rate; OS: overall survival; PPP: platinum-pretreated patients; PFS: progression-free survival; T+: MET ex14 skipping positive in tissue biopsy; TKI: tyrosine kinase inhibitor; TTF: time-to-treatment failure.

### 2.2. CNS Activity

As a result of the specific and potent oncogene inhibition with novel TKIs, leading to improved extracranial disease control and prolonged survival, approximately 20–40% of patients ultimately develop CNS metastases. Brain metastases are associated with poor long-term outcomes, and few therapeutic options are available. Conventional local therapies include whole-brain radiation therapy, rarely curative and burdened by neurocognitive toxicity, and stereotactic radiosurgery, whereas poor performance status and disease burden often preclude neurosurgery. Furthermore, few chemotherapy agents have the ability to cross the blood–brain barrier (BBB), while, in contrast, novel generation TKIs have been specifically engineered to improve their intracranial permeability and activity, eventually leading to better intracranial outcomes.

While first- and second-generation EGFR-TKIs exhibit intracranial efficacy at some level, their concentrations in the cerebrospinal fluid (CSF) only reach 1–5% of the serum concentrations [72,73,74,75,76]. In contrast, osimertinib not only reaches higher intracranial concentrations but also demonstrates substantial intracranial efficacy at its standard daily dose of 80 mg [77,78]. In the FLAURA trial, the median CNS-PFS was not achieved with osimertinib, compared to 13.9 months in patients treated with first-generation TKIs (HR 0.48; 95% CI 0.26–0.86). Additionally, the incidence of CNS progression was lower in the osimertinib arm (6% vs. 15%), regardless of the presence of baseline CNS involvement [20]. Among patients with evaluable brain metastases, the intracranial response rate was substantially higher with osimertinib (91% vs. 68%) [62,79].

In the natural history of *ALK*-translocated NSCLC, most patients develop CNS metastases. First-line alectinib or brigatinib have improved intracranial ORR compared to crizotinib, with intracranial ORRs of approximately 80% and durable intracranial responses [34,35,36]. In cross-study comparisons, ceritinib showed lower CNS penetration, with an intracranial ORR of 72% observed in the ASCEND-4 trial [63]. The third-generation lorlatinib outperformed the intracranial activity of crizotinib in the CROWN trial (83% vs. 23%), with complete responses in 71% of patients treated with lorlatinib. Moreover, the 12-month rate of CNS progression in patients with and without baseline brain metastases was improved in the experimental arm (7% vs. 72% and 1% vs. 18%, respectively) [38,64].

For patients with *ROS*-rearranged NSCLC, the intracranial activity of entrectinib further enforces its use in treatment-naïve patients, with this agent leading to intracranial ORRs of approximately 80%, alongside a median intracranial PFS of 12.0 months and a median intracranial DoR of 12.9 months [44].

Similarly, both tepotinib and capmatinib have shown intracranial activity in patients with *MET* exon-14 skipping, and an intracranial ORR of 68% was observed in treatment-naïve patients treated with capmatinib in the GEOMETRY-Mono-1 study [49,50].

In the LIBRETTO-001 trial, CNS responses with selpercatinib were documented in 85% of patients, regardless of previous systemic treatment and/or radiotherapy, with a median DoR of 9.4 months [71]. Pralsetinib also led to an intracranial ORR of 78% in the ARROW trial [55].

Notably, several targeted agents have shown efficacy in patients with leptomeningeal disease, including lorlatinib and alectinib for ALK-positive cancers and selpercatinib in RET-fusion NSCLC [80,81]. For those with an *EGFR* mutation and leptomeningeal disease, osimertinib has demonstrated significant intracranial activity against brain metastases at a dose of 80 to 160 mg daily [82].

## 3. Novel Generations of Small Molecule Inhibitors in Clinical Development

### 3.1. EGFR

Beyond osimertinib, several third-generation TKIs have been developed, and three of them (lazertinib, almonertinib, furmonertinib) have already been approved with the same indications of osimertinib in Korea and China [83,84,85,86,87]. Table 2 summarizes their principal features, their main clinical results, and, where available, their approval indications.

Similarly to osimertinib, they have proven to be active towards *EGFR*-T790M resistance mutation and showed superiority to 1st or 2nd generation TKI. Moreover, they share the same risk of developing rare and severe toxicity (ILD and QTc prolongation) and are inactive toward *EGFR* p.C797S mutation. In the absence of any head-to-head comparison results, the real advantage taken from more similar drugs available in the market will be the potential improvement in the cost-effectiveness of these drugs [88,89,90].

**Table 2 cancers-15-05079-t002:** 3rd generation EGFR TKIs beyond osimertinib.

	Lazertinib ^a^	Almonertinib ^b^	Furmonertinib ^c^	TY-9591	SH-1028	Limertinib ^d^	Abivertinib ^e^	Befotertinib ^f^	Rezivertinib ^g^
Structure Respect To Osi	pyrimidine and on phenyl rings	cyclopropyl group on the indole group	tphenyl ring and methyl group	Not released	indole ring	Indole and pyrimidine ring	pyrimidine and on phenyl rings	Not released	oxygen replacing on phenyl ring
IC_50_ nM (T790M+)	1.85	0.37	Not released	Not released	0.55	0.3	0.18	Not released	GI_50_ 22 nM
RP2D	240 mg	110 mg	80 mg	160 mg	200 mg	160 mg BID	300 mg BID	75–100 mg	180 mg
MTD	Not reached	Not reached	Not reached	unpublished	unpublished	unpublished	Not reached	Not reached	Not reached
Approved for T790M+	Korea 18 January 2021	China 18 March 2020	China 3 March 2021	-	-	-	-	-	-
Trial	Phase I/II Lee 2020 [84]	Apollo Lu 2020 [91]	Phase I/II Shi 2021 [92]	NCT04204473 Ongoing	Phase I/II Xiong 22 [93]	Phase IIb Li 2022 [94]	Phase I/II Zhou 2022 [95]	Phase I/II Lu 2022 [96]	Phase I Shi 2022 [97]
ORR (T790M+)	58%	69%	74%	-	60.4%	68.8%	56.5%	67.6%	60.5%
mPFS mos (T790M+)	11	12.3	9.6	-	12.6	11	8.5	16.6	9.7
Approved for 1st line	Korea 30 June 2023	China 4 December 2021	China 28 June 2022	-	-	-	-	-	-
Vs 1st TKI	LASER301 Cho 2023 [85]	AENEAS Lu 2022 [98]	FURLONG Shi 2022 [99]	-	NCT04239833 Ongoing	NCT04143607 Ongoing	AEGIS-1 Ongoing	NCT04206072 Lu 2023 [100]	REZOR Ongoing
mPFS (mos)	20.6 vs. 9.7	19.3 vs. 9.9	20.8 vs. 11.1	-	-	-	-	21.1 vs. 13.8	-
ILD	3%	1%	1%	NR	0	NR	NR	2%	NR
Ongoing trials	MARIPOSA Lazertinib vs. Osimertinib vs. Lazertinib/ amivantamab	-	-	FLETEO TY-9591 vs. osimertinib	-	-	-	-	-

Abbreviations: IC_50_ = Half-maximal inhibitory concentration; RP2D = recommended phase 2 dose; ORR = overall response rate; mPFS = median progression-free survival. ^a.^ Lazertinib, also known as (AKA) GNS-1480/YH25448/JNJ-73841937; ^b.^ almonertinib AKA aumolertinib or HS-10296; ^c.^ fulmonertinib AKA Alflutinib/AST2818; ^d.^ Limertinib, also known as (AKA) = ASK120067; ^e.^ abivertinib AKA AC0010/Avitinib/STI-6565; ^f.^ befotertinib AKA D-0316; ^g.^ rezivertinib AKA BPI-7711.

To overcome *EGFR*-C797X mediated resistance, several 4th generation TKIs have been designed and are at difference stages of clinical research (Appendix A) [101]. The first developed drugs (EAI045, JBJ-04-125-02, BLU-701) were rapidly withdrawn since their activity depends on combination with other drugs or lack of efficacy [102,103,104]. In the same line, BLU-945 is also being investigated in combination with osimertinib to improve the activity against *EGFR*-sensitizing mutations [105]. Further, fourth-generation EGFR-TKIs, which have demonstrated proof of activity in cancer models harboring C797S with or without T790M, are being experimented in phase I trials (Appendix A) [106,107,108,109,110,111,112,113,114].

With concern on the *EGFR* and *HER2* Exon 20 insertion mutations that are intrinsically resistant to available EGFR-TKIs, different drugs with similar properties are currently evaluated in clinical research (Appendix A) [115,116,117,118,119,120,121]. The main issue in the development of TKIs targeting *EGFR* or *HER2* exon 20 insertions consists of achieving high selectivity over wild-type receptors in order to increase their therapeutic window [114]. With regard to uncommon *EGFR* mutations, few studies are investigating the efficacy of specific TKIs or combinations: phase II trials of furmonertinib (NCT05548348), sutetinib (NCT05168566) and the afatinib/bevacizumab combination (NCT05267288), phase III trial of almonertinib over standard chemotherapy (NCT04951648).

### 3.2. KRAS

The efficacy of *KRAS*-G12C inhibitors is tempered by the RAS pathway complexity, the concomitant presence of other gene mutations, such as *KEAP1*, and the acquired secondary mutations on the switch pocket II of KRAS [122,123]. New KRAS inhibitors have been designed to inhibit this target more potently and selectively (Table 3 and Appendix A). Among them, divarasib and JDQ433 are on more advanced clinical development. Divarasib (GDC-6036) has been designed to inhibit covalently, selectively, and with more potency *KRAS* G12C compared to sotorasib and adagrasib [124]. JDQ443 has been designed to overcome resistance mechanisms observed with other KRAS G12C inhibitors since it acts through a novel binding mechanism that forms novel interactions with KRAS under the switch II pocket, irreversibly trapping KRAS in the inactive, GDP-bound state reaching the residue C12 without interfering with residue H95 [125].

### 3.3. BRAF and MET

The novel encorafenib and binimetinib combination, already approved for *BRAF*-mutated melanoma, has been investigated in two different phase II trials (ENCO-BRAF, OCEANII) for NSCLC patients. Several trials are experimenting with novel (pan)-RAF inhibitors alone or in combination with MEK, FAK, RAS, or SHP-2 inhibitors in patients with *BRAF*-V600E solid tumors, including NSCLC refractory to BRAK/MEK-inhibitors or harboring other *RAF* alterations (*BRAF* class II and III mutations, *RAF* gene fusions or amplification) [127] (Appendix A).

Despite the development of acquired *MET* mutations seems to be related to the type of MET-inhibitors, at present, no clinical trial has been designed to investigate the re-sensitiveness of these patients to a novel class of MET TKIs, with the exception of a small phase II trial of capmatinib in crizotinib-resistant NSCLC patients, which showed discouraging results [128]. At the state-of-the-art, only the bifunctional anti-EGFR and MET antibody amivantamab have shown modest activity (ORR 17%) in MET-TKI-resistant patients in the Chrysalis trial [129]. Preliminary results were presented from the phase I SHIELD trial of elzovantinib, a MET, SRC, and CSF1R inhibitor, in 52 patients with *MET* dysregulated solid tumors, including 30 patients with *MET*-altered NSCLC (20 *MET-*ex14 skipping mutations, 8 *MET* amplified, 2 other *MET* mutations) [130]. Among the eleven TKI naïve NSCLC patients, the ORR was 36% regardless of dose modifications (Appendix A).

### 3.4. Fusion Genes

Novel 3rd and 4th ALK-TKIs are underway in clinical research. APG-2449 is a novel FAK inhibitor and a third-generation ALK/ROS1-TKI that has shown potent activity towards a range of *ALK*-resistant mutations and brain penetrant capacity in pre-clinical NSCLC tumor models. An ongoing phase I trial is evaluating patients with second-generation TKIs-resistant *ALK/ROS1*-positive NSCLC. Preliminary results have shown an ORR of 28.5% among 14 patients with ALK-TKI refractory NSCLC [131]. TPX-0131 and NVL-655 are the 4th generation ALK-Is. The clinical development of TPX-0131 has been withdrawn due to safety issues; meanwhile, phase I/II of NVL-655 is ongoing [132,133] (Appendix A). Appendix A show the ongoing clinical trials of new *ROS1* and *NTRK* inhibitors (taletrectinib, repotrectinib, NVL-520 among the others for *ROS1* and repotrectinib, VMD-928 and XZP-928 for *NTRK*) with high BBB penetrance and activity towards secondary single or double mutations.

Novel RET inhibitors have been designed to cover acquired resistance mutations while sparing the inhibition of other targets, such as *VEGFR2*, to augment their therapeutic window (Table 4).

## 4. Combination Treatments with New Generation TKIs

### 4.1. EGFR

After the encouraging results of the Japanese phase III trial NEJ009 [140] and phase II OPAL trial [141], the FLAURA-2 trial confirmed the benefit of combining PBC to the third generation TKI osimertinib in 586 treatment naïve *EGFR*+ NSCLC patients, as the combination led to a PFS improvement in 8 months (HR 0.62) [142]. Table 5 depicts trials of the TKI-CT combination for a selected population of patients (p53 mutant or other tumor suppressor genes, lack of circulating DNA clearance, brain metastases). In the post-TKI setting, the COMPEL phase III trial is investigating the role of adding a TKI to standard second-line chemotherapy in order to prevent CNS progression [143].

Histologic transformation into SCLC has been observed in 3–14% of *EGFR*+ NSCLC patients treated with first-generation EGFR-TKIs (gefitinib or erlotinib), frequently mediated by *p53*/*RB1* loss of function [144]. An ongoing phase II trial is evaluating the combination of olaparib and durvalumab in this setting (NCT04538378).

Another important line of research is represented by the development of bispecific antibodies. In the CHRYSALIS study (NCT02609776), the combination of amivantamab plus lazertinib was tested in 20 treatment-naïve Asian patients with *EGFR*-mutant NSCLC, attaining an ORR of 100%. Interestingly, after a median duration of treatment of 33.5 months, median DOR, PFS, and OS have not yet been reached [145].

The phase 3 MARIPOSA study is further investigating the lazertinib/amivantamab combination versus lazertinib or osimertinib alone in 1074 treatment-naïve NSCLC patients with EGFR-common mutations, and positive results have been preliminarily announced [146]. Indeed, in the MARIPOSA2 phase III trial, the combination of amivantamab alone or plus lazertinib with standard PBC has led to a PFS improvement in NSCLC patients harboring common *EGFR*-mutations after failure of treatment with osimertinib [147].

Among the different TKI-TKI combinations, the major interest is focused on MET inhibition. Alongside the first-line combination of savolitinib plus osimertinib (FLOWERS, NCT05163249; NCT04743505), major efforts are being oriented towards the post-3rd generation TKI setting. Based on the results of preliminary trials with gefitinib plus capmatinib and osimertinib plus savolitinib [148], phase II and III trials have been designed to confirm the efficacy of these combinations [149,150]. Moreover, the ORCHARD trial is a biomarker-directed phase II platform study evaluating the optimal treatment for individual patients with *EGFR-*mutant NSCLC [151] (Table 5, Appendix A).

### 4.2. Other Driver Gene Mutations

Appendix A summarizes ongoing TKI-based combinations with chemotherapy, antiangiogenics, multitargeted drugs, or immune-modulating agents for NSCLC patients with *EGFR* or *HER2* exon 20 insertion mutations.

Several combinations have been designed and are under development in clinical trials to overcome resistance to KRAS inhibitors (Appendix A). Indeed, trials have been designed to investigate the MEK-inhibitors and ICI combinations in solid tumors, *BRAF* or *KRAS* mutated NSCLC on the basis of preclinical data and case reports suggesting that MEK inhibition can modulate CTLA-4 expression and potentially increase the efficacy of ICI [152,153] (Appendix A).

Few clinical trials are ongoing with TKI-based combinations of drugs (TKI plus MEK-I or ICI plus amivantamab), aiming to overcome the occurrence of resistance to MET inhibitors (Appendix A).

### 4.3. Fusion Genes

Several trials are experimenting with the feasibility of different ALK, ROS1, or RET TKI-based combinations plus different classes of drug: chemotherapy, antiangiogenesis, multi-targeted kinase inhibitors (lenvatinib, crizotinib, apatinib), or selective inhibitors towards different targets (MEK, MET, or SHP-2), ICI, or immunomodulators (Appendix A).

## 5. The Issue of Sequencing Treatments with New Generation TKIs

In the dynamic landscape of targeted therapy for driver gene alterations in cancer, the introduction of novel generations of TKIs has ushered in a new era of treatment paradigms. These advancements have conferred delayed resistance, prolonged overall survival, and enhanced CNS penetration compared to their predecessors [20,64,154,155]. This transformative impact has often led to the relocation of these novel agents from later lines of therapy to the front-line treatment setting. While these innovations represent a substantial leap forward in cancer care, they raise a significant and somewhat paradoxical challenge: the potential obsolescence of TKI treatment sequencing and the diminished role of previous TKI generations.

However, this progress comes with its own set of complexities, particularly in terms of toxicity profiles. While the safety and tolerability of these novel agents are generally manageable, they often differ from those of their predecessors, requiring new learning curves for medical oncologists on the management of novel and peculiar adverse events in clinical practice (e.g., cognitive effects with lorlatinib and entrectinib, the management of interstitial lung disease occurring at different rates with different drugs) [59,156].

This strategic shift towards the front-line adoption of novel TKIs raises a critical question regarding the sequencing of TKI treatments. Historically, the sequencing of TKIs was a vital aspect of managing drug resistance and optimizing patient outcomes. Patients who developed resistance to an earlier generation of TKI often had the option to switch to a subsequent generation with a different mechanism of action, thereby extending the duration of disease control [18,33]. However, the ascendancy of novel TKIs as first-line therapies has effectively closed this avenue. With the superior efficacy of these agents, previous generations have become less relevant in the treatment algorithm, relegating them to a historical perspective rather than a therapeutic option. As such, whereas TKI sequencing from first- to novel generations relegated chemotherapy options at the end of the treatment sequence, anticipating novel TKI generations in the front-line setting made the role of the ‘old’ chemotherapy options being revived as a necessary second-line treatment option, outside clinical trials [2].

Nonetheless, this paradigm shift underscores the imperative for continuous innovation and adaptation in oncology as the field continues to evolve to meet the ever-changing needs of patients with driver gene alterations.

## 6. The Issue of Resistance: Selective Pressure on Resistant Clones

Resistance to cancer therapies is a formidable challenge in the field of medical oncology, and a crucial aspect of this challenge is the selective pressure imposed on resistant clones within heterogeneous tumors [7,157]. Molecular heterogeneity, a pervasive feature of most cancers, lies at the heart of acquired resistance development. Tumors, even when sharing similar clinical characteristics, are composed of a mosaic of molecular clones, each characterized by unique genetic and phenotypic traits [158]. This inherent diversity within tumors provides fertile ground for the emergence of drug-resistant clones, each endowed with specific survival advantages and resistance mechanisms [7]. Particular subgroups referred to as Drug-Tolerant Persister (DTP) cells, have the capacity to endure high-dose treatments [159]. Intriguingly, these sub-clones possess distinctive stem cell markers and can adapt their characteristics in response to therapy-induced selective pressure [159]. DTPs, along with de novo mutations and preexisting resistance mechanisms, are among the potential strategies employed by cancers to evade the pressures exerted by anticancer drugs [157,158].

The consequence of this molecular diversity is that therapeutic interventions, while initially effective against a subset of tumor cells, inadvertently create an environment conducive to the survival and proliferation of drug-resistant clones. The selective pressure imposed by treatments favors the outgrowth of these resistant populations over time, ultimately leading to therapeutic failure [158]. This selective pressure is a dynamic process in which sensitive tumor cells are progressively eliminated, allowing resistant clones to dominate [160].

The role of druggable driver mutations further underscores the complexity of acquired resistance development. In cases where tumors harbor well-defined driver mutations, resistance mechanisms are often intricately linked to these drivers. However, the potency of the therapeutic agent employed can significantly influence the nature of resistance. High-potency drugs, as novel generations of TKIs are, can exert more substantial selective pressure, potentially driving the development of resistance mechanisms that are independent of the drug’s primary target [157,161].

A notable example is observed in the treatment of *EGFR*-mutant lung cancers, where the use of different generations of tyrosine kinase inhibitors (TKIs) leads to distinct resistance patterns [161]. First-generation TKIs like gefitinib and erlotinib are associated with a high incidence of the p.T790M resistance mutation. In contrast, second-generation TKIs such as afatinib have a reduced incidence of p.T790M mutations, while third-generation TKIs like osimertinib exhibit even lower rates of *EGFR*-dependent resistance mechanisms [23,162]. Instead, under the selective pressure of more potent drugs like osimertinib, alternative resistance mechanisms such as *MET* amplification can become prevalent [163]. Similarly, the different structure and potency among first-, second-, and third-generation ALK TKIs lead to a different selection of resistance mutations [164,165]. In addition, the sequential use of subsequent generations of TKIs may lead to the emergence of compound resistance patterns [166].

Indeed, under the selective pressure of more potent compounds, as novel generations of TKIs, resistant clones emerge, harboring more complex and hard-to-target resistance mechanisms [7,167]. Understanding the interplay between selective pressure, molecular heterogeneity drug potency, binding affinity, and structure is critical for devising effective strategies to overcome drug resistance and improve treatment outcomes in cancer patients.

## 7. Conclusions and Future Directions

From the humble beginnings of the first-generation TKIs, exemplified by erlotinib and gefitinib in *EGFR*-mutant disease, to the cutting-edge third-generation agents such as osimertinib and extending the application of TKI treatment to other oncogene-driven diseases, we have witnessed a remarkable transformation in the therapeutic landscape of NSCLC [168]. These targeted therapies have not only prolonged the lives of countless patients but have also provided a blueprint for precision medicine in oncology.

As we conclude our exploration of the strengths and limitations of novel generations of TKIs in NSCLC, it is evident that these agents have significantly improved disease control and survival rates among patients harboring specific genetic alterations. However, it is equally clear that challenges persist, and there is much work yet to be done to optimize their use and expand treatment options. One notable limitation is the development of resistance mechanisms, which underscores the need for ongoing research into novel therapeutic strategies.

One promising avenue lies in the realm of Antibody-Drug Conjugates (ADCs). These innovative biopharmaceuticals combine the specificity of monoclonal antibodies with the cytotoxic potency of chemotherapy drugs (payload), offering a new approach to target oncogenic pathways in NSCLC [169]. ADCs have the potential to overcome some of the limitations of TKI therapy, particularly in cases of resistance and heterogeneous tumor populations. To date, different ADCs are being investigated in driver-mutant NSCLCs after the failure of standard TKI treatment [170]. In a future perspective, these agents can be designed to target specific driver mutations, either by direct antibodies or even using TKIs as pharmaceutical components (as payload or instead of monoclonal antibody) of the ADCs, providing a level of precision therapy that was previously unthinkable [169].

In addition, combination treatments represent another potential strategy. The intricate biology of NSCLC, characterized by the crosstalk between multiple signaling pathways, necessitates a multifaceted approach to therapy. Combinations of TKIs with chemotherapy agents as recently demonstrated in the FLAURA-2 trial with osimertinib plus platinum-doublet, have shown promise in enhancing antitumor responses and delaying the emergence of resistance [142]. Furthermore, rational combinations of TKIs with other targeted therapies, such as MET inhibitors [171], are being actively investigated to address resistance mechanisms and broaden the scope of effective treatment options.

## Figures and Tables

**Figure 1 cancers-15-05079-f001:**
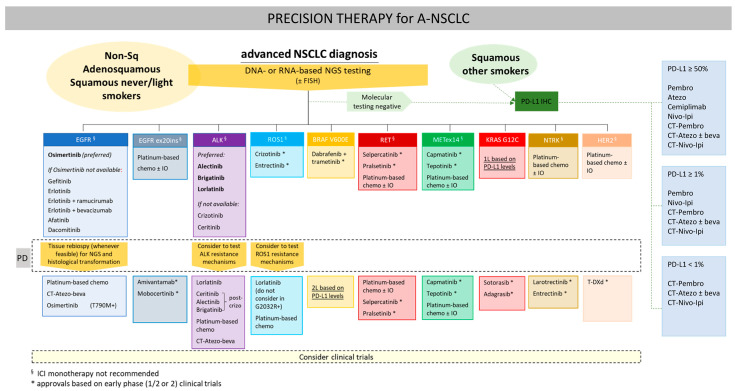
Current treatment options in advanced NSCLC according to molecular gene testing. Novel generations of TKIs, where available, were initially positioned in a therapeutic sequence, but they are established as front-line treatments across driver mutations. Drug classes: Tyrosine kinase inhibitors: Osimertinib, gefitinib, erlotinib, afatinib, dacomitinib, mobocertinib, alectinib, brigatinib, lorlatinib, ceritinib, crizotinib, entrectinib, selpercatinib, pralsetinib, capmatinib, tepotinib, larotrectinib|small molecule inhibitors: dabrafenib, trametinib, sotorasib, adagrasib|bispecific antibodies: amivantamab|Antibody-drug conjugates: trastuzumab-deruxtecan|Abbreviations: CT = chemotherapy; Atezo = atezolizumab; Pembro = pembrolizumab; Nivo = nivolumab|Ipi = ipilimumab|beva = bevacizumab.

**Table 3 cancers-15-05079-t003:** Summary of the novel KRAS inhibitors most advanced in clinical research.

Drug	IC_50_	Clinical Trial(s)	Results	Safety Profile	Ongoing Clinical Trial(s)
Divarasib (GDC-6036) 400 mg OD	0.0029 nM	Phase I/II GO42144 Sacher 2023 [124]	ORR 53.4% mPFS 13.1 months	AE rate 93% G ≥ 3: 12% Most common AEs nausea (74%), diarrhea (61%), and vomiting (58%)	NAUTIKA1 NCT04302025 Biomarker-driven Neoadjuvant platform
JDQ443 200 mg BID	0.012 nM	Phase I–II KontRast-01 Tan 2022 [126] Phase III KontRASt-02 NCT05132075 Ongoing	ORR 57% mDoR 4 months	AE rate 71.8% G ≥ 3: 12.8% Most common AEs fatigue (30.8%) nausea (17.9%) edema (15.4%) diarrhea (12.8%) vomiting (12.8%)	KontRASt-04 JDQ433C12301 1stline JDQ433 + TNO155 vs. CT + ICI STRIDER NCT05999357 Ph II BM + KontRASt-06 NCT05445843 1st line PD-L1 neg or PD-L1+/SKT11+

Abbreviations: IC50 = Half-maximal inhibitory concentration; OD = once daily; BID = two times a day; nM = nanomolar; ORR = overall response rate; mPFS = median progression-free survival; mDoR = median duration of response; AE(s) = adverse event(s); G = grade; Ph = phase; CT = chemotherapy; ICI immune-checkpoint inhibitor; Tx = treatment.

**Table 4 cancers-15-05079-t004:** Summary of clinical results of novel RET inhibitors.

Drug	EC/IC_50_ CCDC6RET Ratio	IC_50_G810R	Clinical Trial	Results	Ongoing Clinical Trial
TPX-0046 [134]	IC_50_ < 10 nM	+ 17 nM	Phase I/II NCT04161391	Terminated (Adverse change in the risk/benefit)	Drug withdrawn
Zeteletinib (BOS172738) 150 mg OD	IC_50_ < 1 nM	Not released	Phase I/II Schöffski 2021 [135]	ORR 33% mDoR not reached	Phase I/II Schöffski 2021
HA121-28 600 mg OD	Data not released	Data not released	Phase I/II Zhao 2021 [136]	Post-CT ORR 41% mPFS not reached	NCT05117658 Ph II trial Post-CT
TAS0953/HM06 [137]	IC_50_ 0.02–s0.1 µM	+	MARGARET Phase I/II NCT04683250	Ongoing	MARGARET Phase I/II NCT04683250
SY-5007 160 or 200 mg BID	IC_50_ < 1 nM	Not released	Phase I/II Zhou 2023 [138]	ORR 75% mDoR not reached	NCT06031558 Ph III trial Single arm
TY-1091 [139]	IC_50_ < 1 nM	++ 9.5 nM	Phase I/II NCT05675605	Ongoing	Phase I/II NCT05675605

Abbreviations: IC50 = Half-maximal inhibitory concentration; OD = once daily; BID = two times a day; nM = nanomolar; ORR = overall response rate; mPFS = median progression-free survival; mDoR = median duration of response; AE(s) = adverse event(s); G = grade; Ph = phase; CT = chemotherapy; ICI immune-checkpoint inhibitor; Tx = treatment.

**Table 5 cancers-15-05079-t005:** Main TKI-based combinations either in tx naïve or previously TKI pretreated EGFR-mutant NSCLC.

	CT	Antiangiogenics	Bispecific Antibodies	ADCs or TKIs
First line Tx	FLAURA-2 [NCT04035486] Osimertinib +/− CT (K-I common) TRIAL HAS RESULT	[NCT05263947] Icotinib + bevacizumab (K-I L858R)	CHRYSALIS [NCT02609776] Lazertinib+ amivantamab (cohort TKI naive)	[NCT05007938] Befotertinib + icotinib
	TOP [NCT04695925] Osimertinib +/− CT (K-I EGFR/p53+)	[NCT04181060] Osimertinib +/− bevacizumab (K-I sensitizing mutations)	OSTARA [NCT05801029] Lazertinib+ amivantamab (K-I common)	METLUNG [NCT05445791] 1st or 2nd TKI +metformin (KI sensitizing mutations)
	[NCT04552613] Standard TKI+/−CT (K-I EGFR/concomitant genes+)	[NCT05507606] Osimertinib +/− bevacizumab (K-I EGFR/p53+)	MARIPOSA [NCT04487080] Lazertinib+ amivantamab (K-I common)	[NCT05880706] Osimertinib+BL-B01D1 (KI common mutations)
	[NCT04410796] Osimertinib +/− CT (K-I ctDNA+ at C2)	[NCT04988607] Osimertinib +/− bevacizumab (K-I L858R)		
	PACE-Lung [NCT05281406] Osimertinib +CT (K-I ctDNA+ at wk3)	AUTOMAN [NCT04770688] Osimertinib + anlotinib (KI common mutations)		
	[NCT05209256] Furmonertinib+/− CT (K-I sensitizing mutations)	[NCT03909334] Osimertinib+/− ramucirumab (KI common mutations)		
	[NCT04923906] Almonertinib+/− CT (K-I sensitizing mutations)	FOCUS-A [NCT04895930] furmonertinib+anlotinib (KI common EGFR)		
	ACROSS1 [NCT04500704] Almonertinib+/− CT (K-I common mutations)	[NCT05271916] Dacomitinib+anlotinib (KI phI common; Ph II L858R)		
	ACROSS2 [NCT04500717] Almonertinib+/− CT (K-I common mutations/Suppressor Genes+)	BELLA [NCT04575415] Bevacizumab + EGFRTKIs (observational study)		
	[NCT03992885] Icotinib + CT (KI sensitizing mutations)	[NCT03904823] Almonertinib + famitinib (K-I sensitizing mutations)		
		[NCT05778149] Almonertinib + anlotinib (K-I common mutations/p53)		
		ACTIVE/CTONG1706 [NCT02824458] Gefitinib +/− apatinib (KI common mutations)		
*MET*-based				FLOWERS (NCT05163249) osimertinib+/− savolitinib (K-I sensitizing/MET+°)
				NCT04743505 Osimertinib +/− savolitinib (K-I sensitizing)
Post-3rd gen EGFR TKI			CHRYSALIS [NCT02609776] Lazertinib+ amivantamab (cohort post-TKIs)	Lung-MAP Sub-Study [NCT05642572] Osimertinib+capmatinib +/− ramucirumab (K-I sensitizing MET AMP)
	CHRYSALIS 2 NCT04077463 Lazertinib+ amivantamab+/− CT (cohort post-osimertinib)		CHRYSALIS 2 NCT04077463 Lazertinib+ amivantamab+/− CT (cohort post-osimertinib)	INSIGHT 2 [NCT03940703] Tepotinib +osimertinib (K-I common/MET+ç)
	MARIPOSA-2 [NCT04988295] CT+/+ amivantamab +/− lazertinib (KI common postosimertinib)		MARIPOSA-2 [NCT04988295] CT+/+amivantamab+/lazertinib (KI common postosimertinib)	SAVANNAH (NCT03778229) Savolitinib+/-osimertinib (K-I common/MET+§)
			PALOMA [NCT04606381] Sc amivantamab (KI solid tumors Common EGFR NSCLC post-TKIs)	SACHI [NCT05015608] CT vs. Osimertinib+ savolitinib (K-I common/MET+@)
	SAFFRON [NCT05261399] CT vs. osimertinib+savolitinib (K-I common/MET+§)		PALOMA2 [NCT05498428] Sc Amivantamab+several regimens (KI Solid Tumors Including txnaive or POSTTKIs common EGFR EGFRex20ins tx naïve)	SAFFRON [NCT05261399] CT vs. osimertinib+savolitinib (K-I common/MET+§)
			PALOMA-3 [NCT05388669] Lazertinib + sc vs. ev amivantamab (k-I common post CT and osimertinib)	[NCT05821933] Furmonertinib+RC108 +/- Toripalimab (k-I sensitizing/MET OE post-TKIs)
		AMAZE-lung [NCT05601973] Lazertinib amivantamab bevacizumab (KI post osimertinib or Lazertinib).	AMAZE-lung [NCT05601973] Lazertinib amivantamab bevacizumab (KI post osimertinib or Lazertinib).	
			[NCT04965090] Amivantamab/lazertinib (KI common after3rdTKI and BM+)	
			PolyDamas [NCT05908734] amivantamab+cetrilumab (KI post osimertinib/CT)	
			NCT03797391 EMB-01 (KI EGFR or MET+)	
			[NCT05498389] EMB-01+ osimertinib (KI postTKIs)	
			[NCT04868877] MCLA-129+osimertinib (k-I NSCLC/solid tumours)	

Abbreviations: CT = chemotherapy; ADC(s) = Antibody(ies) drug conjugated; TKI(s) = Tyrosine kinase inhibitor(s); Tx = treatment; KI = key inclusion criteria; Ph = phase; C = cycle; wk = week; ctDNA = circulating DNA; NSCLC = Non-Small-Cell Lung Cancer; EGFR Epidermal Growth Factor Receptor; EGFREx20ins = EGFR eson 20 insertions; MET OE = MET overexpressed; ç MET+ = gene copy number ≥ 5 and/or MET/CEP7 ≥ 2; § MET+ = MET-Overexpressed (IHC90+) and/or Amplified (GCN≥ 10); @ MET+ = GCN not reported cutoff; ° MET+ = IHC 3+ in ≥75% of tumor cells; MET gene copy ≥ 5 or MET/CEP7 ratio ≥ 2; HER 2 AMP = Her2 amplified; EGFREx20ins = EGFR exon 20 insertions; BL-B01D1 is a bifunctional antibody anti-EGFR/HER3 conjugated to a topoisomerase-I; Famitinib is a multitargeted agent which inhibits stem cell factor receptor (c-Kit; SCFR), vascular endothelial growth factor receptor (VEGFR) 2 and 3, platelet-derived growth factor receptor (PDGFR) and FMS-like tyrosine kinases Flt1 and Flt3; RC108 = ADC anti MET; MCLA-129 is a Human Anti-EGFR and Anti-c-MET Bispecific Antibody.

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
