# Peer review of "New Generations of Tyrosine Kinase Inhibitors in Treating NSCLC with Oncogene Addiction: Strengths and Limitations"

_cancers, 2023, doi:10.3390/cancers15205079_

Round 1

Reviewer 1 Report

Journal of Cancers

Review Article;

The article entitled “New Generations of Tyrosine Kinase Inhibitors in Treating NSCLC with Oncogene Addiction: Strengths and Limitations’’. The authors tried their best to study Tyrosine kinase inhibitors revolutionized the treatment of patients with advanced or metastatic non-small cell lung cancer. Research rapidly moved to the development of newer, more selective, generations of Tyrosine kinase inhibitors, obtaining improved results in terms of disease control and survival. The use of novel generations of Tyrosine kinase inhibitors is not without limitations. We reviewed the main results obtained, as well as the clinical trials ongoing, with Tyrosine kinase inhibitors in oncogene-addicted non-small cell lung cancer, together with the biology underlying their potential strengths and limitations. Novel generations of Tyrosine kinase inhibitors allowed delayed resistance, prolonged survival and improved brain penetration compared to previous generations, although with different toxicity profiles, that generally moved their use from further lines to the front-line treatment. Under the selective pressure of such more potent drugs, resistant clones emerge harboring more complex and hard-to-target resistance mechanisms.

Comments for Authors

Ø  The author needs to include keywords.

Ø  The author needs to include the latest reference in the introduction section.

Ø  Could the author include the natural agent's role as TKIs?

Ø  Include the full-size image in Figure 1.

Ø  As the TKIs are used as combinational therapy, the author didn’t discuss the adverse effects in relapse cancer patients.

Ø  The author needs to revise the MOA in the figure.

Ø  Check grammar and spelling throughout the manuscript. There are some mistakes.

Cite the following references;

·         DOI: 10.2174/1871520622666220831124321

·         DOI: 10.1038/s41419-021-03771-z

Author Response

Ø  The author needs to include keywords. We thank the reviewer for pointing out this: we included keywords as required

Ø  The author needs to include the latest reference in the introduction section. References have been updated

Ø  Could the author include the natural agent's role as TKIs? We thank the reviewer for this comment, though this is out of the scope of our review. however, we better discriminated the specific Moa of small molecule inhibitors where not intended to be TKIs

Ø  Include the full-size image in Figure 1. Full-size image has been uploaded

Ø  As the TKIs are used as combinational therapy, the author didn’t discuss the adverse effects in relapse cancer patients. We thank the reviewer for this comment, however this is out of the scope of our review.

Ø  The author needs to revise the MOA in the figure. We thank the reviewer for this suggestion: we included specific details in the figure legend

Ø  Check grammar and spelling throughout the manuscript. There are some mistakes. We thank the reviewer: we double-checked the manuscript and corrected grammar and spelling where needed.

Reviewer 2 Report

This manuscript falls under cancers scope and presents a comprehensive review on research titled “New generations of tyrosine kinase inhibitors in treating NSCLC with oncogene addiction: strengths and limitations”. The manuscript consists of 26 pages, 1 figure and 5 tables. The MS reviewed a variety of reliable methodology on tyrosine kinase inhibitors (TKIs) in treating advanced or metastatic non-small cell lung cancer (NSCLC). The topic original and relevant in the field of study. The Abstract provides the highlights of the key contents of the main text. The Introduction provides enough background information to justify the study. The MS adequately described up to date findings and the conclusion consistent with the evidence from published studies. Researchers concluded that TKIs agents have significantly improved disease control and survival rates among patients, however challenges persist, particularly with the limitation of developing resistance. The references are appropriate and relevant to the research. However, minor typographical and grammatical errors need addressing.

Minor typographical and grammatical errors need addressing.

Author Response

We thank the reviewer. We double-checked the manuscript and corrected the minor typographical and grammatical errors

Reviewer 3 Report

An exciting and carefully prepared manuscript regarding the possibilities of NSCLC therapy. The authors effectively summarized what is known about TKIs in oncogene-addicted NSCLC. The manuscript deserves publication and will undoubtedly attract a large readership, given the importance of the topic and the thoroughness of the manuscript.

Below are my comments on the manuscript:

1. there is no simple summary in the content of the paper,

2. the citation method does not comply with the editorial requirements - this is only a minor remark and of course, I believe that the authors will adapt this citation method to the MDPI requirements,

3. Fig. 1 is very valuable and informative, but its graphic quality is poor,

4. authors quote websites - this is allowed, but please provide the date of access,

5. moreover, the manuscript is very well written - I congratulate the authors,

6. references should be extended to include the following literature:

https://www.mdpi.com/1422-0067/22/21/11659

https://www.mdpi.com/2072-6694/14/12/2940

https://www.mdpi.com/2072-6694/14/24/6061

https://www.mdpi.com/2072-6694/12/3/731

Author Response

1. there is no simple summary in the content of the paper. We thank the reviewer. We provided the simple summary in the revised version.

2. the citation method does not comply with the editorial requirements - this is only a minor remark and of course, I believe that the authors will adapt this citation method to the MDPI requirements. We thank the reviewer, we updated the citation method as required

3. Fig. 1 is very valuable and informative, but its graphic quality is poor. We thank the reviewer, we uploaded full-size version of Fig1 as separate file

4. authors quote websites - this is allowed, but please provide the date of access. We thank the reviewer, we updated the access date

5. moreover, the manuscript is very well written - I congratulate the authors. That you for this comment

6. references should be extended to include the following literature: we included the suggested references

Round 2

Reviewer 3 Report

The authors have satisfactorily addressed all of my concerns.